# Multi-layer Stack Ensembles for Time Series Forecasting

**Nathanael Bosch**[1,*]  **Oleksandr Shchur**[2]  **Nick Erickson**[2]  **Michael Bohlke-Schneider**[2]
**Caner Türkmen**[2]

[1]Tübingen AI Center, University of Tübingen
[2]AWS
[*]Work done during an internship at AWS

**Abstract**  Ensembling is a powerful technique for improving the accuracy of machine learning models, with methods like stacking achieving strong results in tabular tasks. In time series forecasting, however, ensemble methods remain underutilized, with simple linear combinations still considered state-of-the-art. In this paper, we systematically explore ensembling strategies for time series forecasting. We evaluate 33 ensemble models—both existing and novel—across 50 real-world datasets. Our results show that stacking consistently improves accuracy, though no single stacker performs best across all tasks. To address this, we propose a multi-layer stacking framework for time series forecasting, an approach that combines the strengths of different stacker models. We demonstrate that this method consistently provides superior accuracy across diverse forecasting scenarios. Our findings highlight the potential of stacking-based methods to improve AutoML systems for time series forecasting.

## 1 Introduction

Time series forecasting plays a critical role in applications ranging from inventory management (Croston, 1972) to energy systems (Hong et al., 2016) and public health (Tsang et al., 2024). Driven by growing interest in the machine learning (ML) community, numerous forecasting models have been proposed in recent years. Yet no single model is universally dominant. Large-scale benchmarks show that while some models perform better on average, none consistently outperforms all others across diverse datasets and tasks (Aksu et al., 2024; Godahewa et al., 2021).

This variability in model performance motivates the use of automated machine learning (AutoML) for time series forecasting. By evaluating multiple models on a given dataset, an AutoML system can select the one that performs best for the specific forecasting task at hand. However, relying solely on model selection is often insufficient to achieve the lowest possible forecast error.

A growing body of work—both in academic research and prediction competitions—demonstrates that combining forecasts from multiple models can lead to substantial improvements in predictive performance (Bojer and Meldgaard, 2021; Makridakis et al., 2018, 2022). Yet, several key questions remain unanswered. While many forecast combination methods have been proposed over the years (X. Wang et al., 2023), there is limited understanding of which ensembling strategies are most effective in practice. In the related field of tabular AutoML, techniques such as multi-layer stacking repeatedly demonstrate state-of-the-art performance (Gijsbers et al., 2024). In contrast, time series forecasting has seen limited adoption of such techniques, with most practical systems relying on simple linear combinations of model outputs (Herzen et al., 2022; Shchur et al., 2023). Moreover, several studies suggest that in forecasting, simple arithmetic averages outperform more sophisticated ensembling techniques, which is often referred to as the "forecast combination puzzle" (Smith and Wallis, 2009; Stock and Watson, 2004).

To address this gap, we conduct a large-scale empirical study of forecast combination methods, aiming to identify strategies with the best predictive performance. Our main contributions are:

- **Empirical evaluation of ensembling methods**. We conduct a large-scale benchmark of 33 forecast combination methods across 50 real-world datasets on point and probabilistic forecasting tasks. Our results show that ensembling consistently improves the predictive accuracy, with learning-based ensembles significantly outperforming simple aggregation strategies.

- **Multi-layer stacking framework**. We propose a framework for building multi-layer stack ensembles for time series forecasting, designed to combine the strengths of different ensembling methods. This approach consistently outperforms any individual ensemble across both point and probabilistic forecasting tasks. Additional ablation studies reveal when and why multi-layer stacking is effective, demonstrating its adaptability and robustness across datasets.

## 2 Background

### 2.1 Probabilistic time series forecasting

Given a collection of $N$ time series $\mathcal{D} = \{y_{i,1:T}\}_{i=1}^N$, with $y_{i,t} \in \mathbb{R}$, the goal of time series forecasting is to predict the future $H$ values $y_{i,T+1:T+H}$ for each time series $i = 1, \ldots, N$. Specifically, in *probabilistic* time series forecasting, the aim is to model the conditional distribution $p(y_{i,T+1:T+H} \mid y_{i,1:T})$ for all $i$. Often, it is sufficient to produce a quantile forecast rather than the full distribution. Given a predefined set of quantile levels $\mathcal{Q} \subset (0,1)$ with $|\mathcal{Q}| = Q$, the goal is to predict $\hat{y}_{i,T+h}^q \in \mathbb{R}$ such that $P(y_{i,T_i+h} < \hat{y}_{i,T_i+h}^q) = q$, for all $q, i, h$. Alternatively, if uncertainty quantification is not needed, a conditional mean or median forecast may suffice. In this work, we primarily focus on quantile forecasts, but all methods also apply to point forecasts and we will also evaluate these in Sec. 6.

Time series forecasting *models* $f_i : \mathbb{R}^T \to \mathbb{R}^{H \times Q}$ produce these quantile predictions, mapping $f_i : y_{i,1:T} \mapsto \hat{y}_{i,T+h}^q$ for all $i$. Here, the collection of models $\{f_i\}$ can be individually learned for each time series, an approach known as *local* time series modeling (Januschowski et al., 2020). Local time series models include *seasonal naive*, *exponential smoothing*, and *ARIMA*, which typically fit a small set of model parameters on each item individually, in order to capture simple patterns in the data such as trends and seasonality (R. Hyndman and Athanasopoulos, 2021).

On the other hand, a *global* time series model uses a single model for all time series, that is $f_i = f$ for all $i$. The function $f$ is learned over the entire training set $\mathcal{D}$. The aim in global time series modeling is to use a higher capacity function $f$ (e.g., a neural network) to capture both simple temporal patterns and any dynamics common to all time series in a dataset. This approach includes many recent deep learning-based approaches, such as *DeepAR* (Salinas, Flunkert, et al., 2020), *TFT* (Lim et al., 2021), *PatchTST* (Nie et al., 2022), *DLinear* (Zeng et al., 2023), or *TiDE* (Das et al., 2023).

Finally, *pretrained* models also fit parameterized functions to data, but are trained on large corpora of real-world and/or synthetic time series from diverse domains. The model is then applied as a zero-shot forecaster, that is, without any additional training on the dataset $\mathcal{D}$ associated with the given forecasting task. Recent examples of such pretrained models include *Chronos* (Ansari et al., 2024), *TimesFM* (Das et al., 2024), *Moirai* (Woo et al., 2024), and *TTM* (Ekambaram et al., 2024).

Time series models are trained via minimizing loss function $\mathcal{L} : \mathbb{R}^{H \times Q} \times \mathbb{R}^H \to \mathbb{R}_{\geq 0}$ over the training set. Examples include quantile loss for probabilistic forecasting, or mean absolute error for point forecasting (R. Hyndman and Athanasopoulos, 2021).

### 2.2 Forecast combination

While many approaches to modeling exist, none of these dominate others in terms of accuracy or efficiency. The best forecasting approach is therefore often a combination of different models (X. Wang et al., 2023). Let $f_{i,m}$ denote the forecast model for item $i$, obtained by using the modeling approach $m \in 1, \cdots, M$. Forecasts from these models can be combined with a combination function $g_i : (\mathbb{R}^{H \times Q})^M \to \mathbb{R}^{H \times Q}$ that operates on the outputs of a collection of base forecasting models

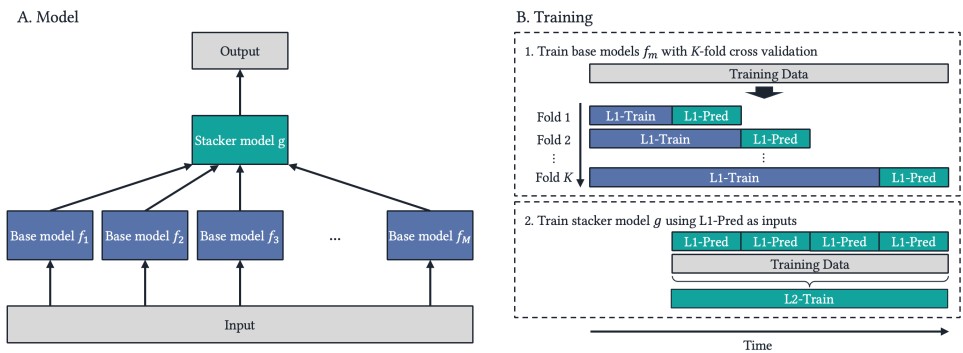

Figure 1: Architecture and training procedure of a single-layer stacker model.

$\{f_{i,m}\}$ to produce a single aggregated forecast. Composing the combination functions $g_i$ with base forecasters $\{f_{i,m}\}$, we obtain forecast ensembles $\overline{f}_i$

$$\overline{f}_i : \mathbb{R}^T \to \mathbb{R}^{H \times Q}, \quad y_{i,1:T} \mapsto g_i(f_{i,1}(y_{i,1:T}), f_{i,2}(y_{i,1:T}), \ldots, f_{i,M}(y_{i,1:T})), \tag{1}$$

for all $i$. See also Fig. 1 for a visual description of the approach.

Similar to base models, combination methods can also be *global* with $g_i = g$ for all $i$. This is possible regardless of whether the base models are local or global. For example, we could learn global combinations $g$ of individually fitted local models $\{f_{i,m}\}$. Conversely, we could use a different combination method $g_i$ for each time series, when combining global forecasting methods $\{f_m\}$. Combinations of global and local models as base models are also possible. In the following, we will rely on this generality to drop the item index $i$ from our notation. For example, in the following $\hat{y}_m$ may denote either the predictions for a single item $\hat{y}_{i,m}$ or for all items $\{y_{i,m}\}_{i=1}^N$, depending on the context.

Likely the most popular forecast combination approach is **simple averaging** via the mean or median of individual forecasts, e.g., $g(\hat{y}_1, \cdots, \hat{y}_M) = \frac{1}{M}\sum_m \hat{y}_m$. Simple averaging has been shown to often outperform individual forecasting models (Perrone and Cooper, 1995; X. Wang et al., 2023). **Model selection** can also be cast in the same notation. That is, $g(\hat{y}_1, \ldots, \hat{y}_M) = \hat{y}_{m^*}$, where $m^*$ is the index of the model with the best validation score, i.e. $m^* := \arg\min_{m=1,\ldots,M} \mathcal{L}(\hat{y}_m, y_{\text{out}})$, where $y_{\text{out}}$ denotes the ground truth future values corresponding to the predictions $\hat{y}_m$.

In the AutoML literature, methods that combine outputs of individual models are referred to as *post-hoc ensembles* (Purucker and Beel, 2023) to differentiate them from other ensembling techniques like bagging and boosting. In forecasting literature, apart from *forecast combinations* (Clemen, 1989), these methods are called forecast ensembles (Adhikari and Agrawal, 2012) and model averaging (Montero-Manso et al., 2020). We use these terms interchangeably throughout the paper.

## 3 Stacking for time series forecasting

### 3.1 Stacking

Simple averaging and model selection are strong baseline approaches. In this work, however, we explore a broader class of methods that can *learn* the combination functions $g$ from data. In ML literature, such methods that learn to combine the outputs of other models are often called *stacking* methods (Zhou, 2012, Ch. 4). We now review some model families that fall into this category.

**Model averaging**. One generalization of simple averaging is weighted model averaging, with a stacker model of the form $g(\hat{y}_1, \ldots, \hat{y}_M) = \sum_{m=1}^M \omega_m \hat{y}_m$, with weights $\omega \in \mathbb{R}^M$. The weights $\omega$ can be chosen based on the **model performance**. For example, the weights can be computed as $\omega_m \propto h(\mathcal{L}(\hat{y}_m, y_{\text{out}}))$, where $h$ are chosen as $h_{\text{inv}}(L) = 1/L$, $h_{\text{sqr}}(L) = 1/L^2$, or $h_{\text{exp}}(L) = \exp(1/L)$, subject to $\sum_{m=1}^M \omega_m = 1$ (Pawlikowski and Chorowska, 2020).

Alternatively, stacker weights $\omega$ can be learned from data by minimizing the loss function, $\omega = \arg\min_{\omega' \in \mathbb{R}^M} \mathcal{L}(g(\hat{y}_1, \ldots, \hat{y}_M), y_{\text{out}})$, often subject to $\omega_m \geq 0$ and $\sum_m \omega_m = 1$. If the loss is differentiable, weights can be learned via a **numerical optimization** method such as gradient descent. Another popular approach is the greedy **ensemble selection** algorithm (Caruana et al., 2004), which adds models to the ensemble with replacement. Ensemble selection is used by default in several AutoML forecasting frameworks (Deng et al., 2022; Shchur et al., 2023; Zöller et al., 2024).

**General combinations via linear regression.** Instead of assigning a single weight per model, weights can vary with item $i$, time step $h$, quantile $q$, or any combination thereof. Reintroducing the item index $i$, we write:

$$[g_i(\hat{y}_{i,1}, \ldots, \hat{y}_{i,M})]_{h,q} = \sum_{m=1}^{M} \omega_{i,h,q,m} \cdot [\hat{y}_{i,m}]_{h,q} \tag{2}$$

with weights $\omega \in \mathbb{R}^{N \times H \times Q \times M}$. These weights can be tied across horizon steps, e.g., $\omega_{i,h,q,m} = \omega_{i,q,m}$, or even shared across items. This defines a broad family of linear combinations, as explored by Hasson et al. (2023) through different parameter-tying and regularization schemes. Our evaluation in Sec. 6 includes several of these linear variants.

**Nonlinear regression.** Finally, $g$ can be a nonlinear function, often implemented using tree-based models such as gradient boosting—e.g., as in `darts` (Herzen et al., 2022), `sktime` (Löning et al., 2019), or Gastinger et al. (2021). As with linear models, these can vary in how they tie parameters across items, quantiles, or forecast horizons. Input normalization (e.g., scaling $\hat{y}_m$) is often beneficial before applying nonlinear models. In Sec. 6, we evaluate several nonlinear tabular models.

## 3.2 Training stacker models with time series cross-validation

One important detail that we have not yet addressed is how to obtain the training data for the stacker models. It is conventional wisdom in ensemble training that in order to prevent overfitting, stacker models $g$ should be trained on *out-of-fold* data—i.e., data that was held out during the training of the base models $\{f_m\}$ (Zhou, 2012). Holding out data, in turn, requires special treatment in the case of time series (R. Hyndman and Athanasopoulos, 2021, Section 5.10). In line with this, we use a time series $K$-fold cross validation scheme which works as follows. For each fold $k = 1, \ldots, K$, we first remove the last $j := (K - k + 1)$ windows of size $H$ from each time series, obtaining the training set $y_{1:T-jH}$ used to train the base models $\{f_m\}$. Next, we make $H$-step-ahead predictions with all trained base models, obtaining $M$ quantile forecasts $\{\hat{y}_m^k\}$, each covering the validation window $y^k = y_{T-jH:T-(j-1)H}$. After repeating this process $K$ times, we collect the training set for the stacker model $g$ consisting of the inputs (the out-of-fold predictions $\hat{y}_m^k$ for each model $m$ and fold $k$), and the targets (the out-of-fold data $y^k$ for each fold $k$). See Fig. 1 for a visual depiction.

# 4 Multi-layer stacking for time series forecasting

## 4.1 Multi-layer stacking

As we will see in Sec. 6, the performance of different stacker models can vary greatly across datasets. To address this issue, we propose a *multi-layer stacking* approach that aggregates the outputs of multiple stackers with yet another stacker model (Fig. 2). This approach removes the need to decide on a single stacker model $g$ in advance—we can instead learn the optimal combination of multiple stacker models $\{g_c\}_{c=1}^{C}$, similar to how we considered a set of base forecasting models $\{f_m\}_{m=1}^{M}$. Finally, we can combine the outputs of $\{g_c\}_{c=1}^{C}$ using a new stacker model $s: (\mathbb{R}^{H \times Q})^C \to \mathbb{R}^{H \times Q}$. In the following, we refer to the base models $\{f_m\}_{m=1}^{M}$ as the first layer of the multi-layer stack ensemble, the stacker models $\{g_c\}_{c=1}^{C}$ as the second layer, and the aggregator model $s$ as the third layer; or in short as L1, L2, and L3 models (*cf.* Fig. 2).

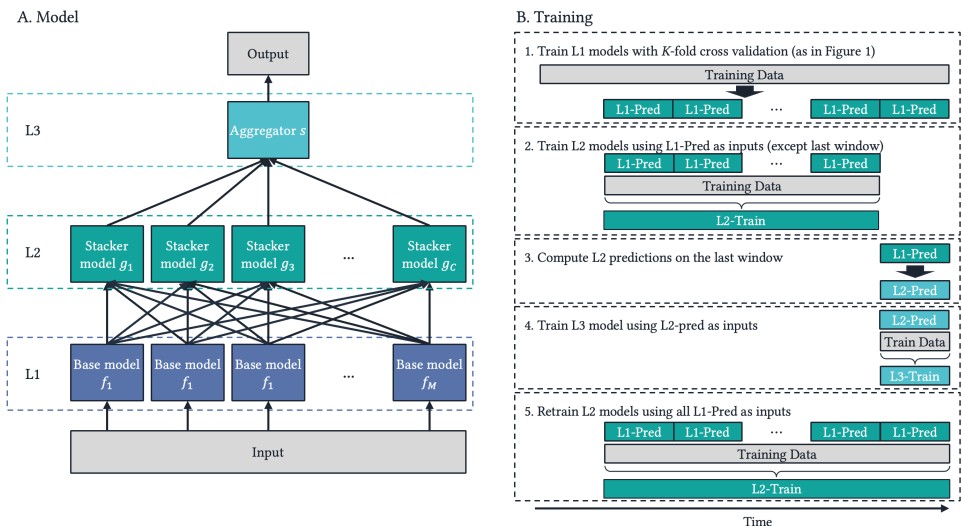

Figure 2: Architecture and training procedure of a multi-layer stacker model.

Concretely, to generate quantile predictions for an item, its past data $y_{i,1:T}$ is used to produce forecasts $\hat{y}_{i,1}, \ldots, \hat{y}_{i,M}$ from each L1 model. These are then aggregated by a composition of L2 models and a single L3 model $s$. The resulting *multi-layer stack ensemble* defines the mapping:

$$y_{i,1:T} \mapsto s(g_1(\hat{y}_{i,1}, \ldots, \hat{y}_{i,M}), \ldots, g_C(\hat{y}_{i,1}, \ldots, \hat{y}_{i,M})).$$

Note that all forecast combination and stacker models $g$ discussed in the previous sections can also be used as the aggregator model $s$. For example, the aggregator $s$ can perform model selection among the L2 stacker models, or it could combine the L2 models into a weighted ensemble.

### 4.2 Training of multi-layer stacker models with two-level time series cross-validation

Similar to the individual stacker models $g$, we need to make sure that the L3 aggregator model $s$ is trained on out-of-fold predictions of the L2 models to avoid overfitting. To achieve this, we adjust the procedure described in Sec. 3.2. Like before, we train the L1 base models using $K$-fold time series cross validation. Next, we only train the L2 stacker models $\{g_c\}$ on the first $K - 1$ validation windows. We then make predictions with the L2 models for the $K$th validation window, and use these predictions together with the ground truth values to fit the L3 aggregator model $s$. Finally, we re-train the L2 models using all $K$ validation windows to ensure that these models have access to the most recent data. The entire process is shown in Fig. 2.

## 5 Related work

**Time series forecasting**. Time series modeling spans classical statistical methods (Box et al., 1970; R. Hyndman, Koehler, et al., 2008), deep learning models (Benidis et al., 2022; Challu et al., 2023; Nie et al., 2022; Salinas, Flunkert, et al., 2020), and more recently, pretrained models (Ansari et al., 2024; Das et al., 2024; Woo et al., 2024). Despite substantial progress, no single model consistently outperforms others across all datasets and problem settings (Aksu et al., 2024; Godahewa et al., 2021). This motivates AutoML systems that automatically train, tune, and combine models to achieve the best performance for a given task (Ali, 2020; Deng et al., 2022; Shchur et al., 2023; Zöller et al., 2024). Our stacking framework is compatible with arbitrary forecasting models, making it complementary to ongoing advances in model development.

**Forecast combinations**. The idea of combining forecasts dates back to Bates and Granger (1969) and has inspired many methods (X. Wang et al., 2023), often featured in winning solutions of prediction competitions (Bojer and Meldgaard, 2021; Makridakis et al., 2018, 2022). Early studies found that simple averages often outperform more complex methods—a phenomenon known as the "forecast combination puzzle" (Smith and Wallis, 2009; Stock and Watson, 2004). However, these results were based on small datasets, statistical models, and point forecasts (Gastinger et al., 2021). Later work showed that more sophisticated methods outperform simple averages when applied to modern ML models, larger datasets, and probabilistic forecasting (Hasson et al., 2023). Our results in Sec. 6 advance this discussion and show that stacking improves over simple aggregation strategies for both point and probabilistic forecasting.

**Stacking**. Stacking, or stacked generalization, was introduced by Wolpert (1992) and Breiman (1996), where a single stacker model combines predictions from base models to produce the final output (Van der Laan et al., 2007). Variants of stacking have been studied extensively, including for quantile regression (Fakoor et al., 2023). In AutoML systems, a common approach is ensemble selection (Caruana et al., 2004), popularized by Auto-Sklearn (Feurer et al., 2015). Multi-layer stack ensembles first appeared in competition-winning solutions (Itericz and Semenov, 2016; Koren, 2009), and gained broader adoption with AutoGluon-Tabular (Erickson et al., 2020), which introduced a robust bagging-based implementation. However, existing work on multi-layer stacking is limited to tabular tasks. Our paper extends this framework to point and probabilistic time series forecasting.

## 6 Experiments

The main goal of our experimental analysis is to determine which forecast combination methods result in the best prediction accuracy (Sec. 6.1). In subsequent experiments, we aim to get better understanding of these results. For this purpose, we investigate the effects of various design choices such as selection of L1 models or the amount of validation data (Sec. 6.2–6.5).

### 6.1 Which combination methods produce the most accurate forecasts?

In our main experiment, we perform a large-scale benchmark comparison of 33 forecast combination methods, including the single-layer (Sec. 3) and multi-layer stacking (Sec. 4).

**Datasets**. We use 50 univariate datasets from Ansari et al. (2024) and Woo et al. (2024), covering diverse domains and frequencies, totaling 90K time series with 110M observations (see Tab. 5). To ensure that enough data is available for training base and stacker models, we only keep time series with at least $8 \times H$ observations (where $H$ is the forecast horizon) in each dataset. As this filtering changes the data compared to the original publications, we re-evaluate all models ourselves.

**Models**. We consider 11 base forecasters (**L1 models**) covering all popular model categories: statistical models (*SeasonalNaive*, *AutoETS*, *Theta*), deep learning models (*DeepAR*, *PatchTST*, *TFT*, *TiDE*, *DLinear*), gradient-boosted trees (*DirectTabular*, *RecursiveTabular*), and one pretrained model (*Chronos-Bolt*). We use model implementations from AutoGluon–TimeSeries (Shchur et al., 2023), trained with default hyperparameters until convergence, without hyperparameter tuning for simplicity. These 11 L1 models are used as input to all the subsequent combination methods.

In addition, we train 31 combination methods (**L2 models**) from Sec. 3. These include 2 *simple averages*, *model selection*, 3 *performance-weighted averages*, 3 *greedy ensembles*, 18 *linear models*, and 4 *nonlinear models*. A detailed breakdown for each model category is available in App. B.

Finally, we consider 2 multi-layer stacking approaches (**L3 models**) from Sec. 4: *stacker model selection* that selects the best L2 model based on the performance on the last validation window, and *multi-layer stacking* that fits a greedy ensemble on top of the L2 models. To keep the training time reasonable, we limit the second level of the ensemble to 14 a priori chosen L2 models.

Table 1: Aggregated probabilistic forecasting performance of the representative combination methods based on SQL. Best result in **bold**, second best underlined. Individual results for all methods and datasets are available in Tab. 8

| Method | (↑) Elo | (↑) Champion | (↓) Average rank | (↓) Average relative error | (↓) Median marginal training time |
|---|---|---|---|---|---|
| Median | 1000 | 3 | 5.92 | 1.000 | **1s** |
| Model selection | 1049 | 9 | 5.43 | 1.001 | 2s |
| Performance-based average | 1130 | 3 | 4.60 | 0.963 | 2s |
| Greedy ensemble selection | 1191 | 4 | 3.92 | 0.952 | 11s |
| Linear model | 1220 | 3 | 3.62 | 0.947 | 11s |
| Nonlinear model | 1046 | 4 | 5.43 | 1.011 | 27s |
| Stacker model selection | 1157 | 1 | 4.27 | 0.973 | 484s |
| Multi-layer stacking | **1306** | **20** | **2.81** | **0.945** | 721s |

Table 2: Aggregated probabilistic forecasting performance of the representative combination methods based on MASE. Best result in **bold**, second best underlined. Individual results for all methods and datasets are available in Tab. 9.

| Method | (↑) Elo | (↑) Champion | (↓) Average rank | (↓) Average relative error | (↓) Median marginal training time |
|---|---|---|---|---|---|
| Median | 1000 | 7 | 5.54 | 1.000 | **1s** |
| Model selection | 997 | 8 | 5.58 | 1.026 | **1s** |
| Performance-based average | 1059 | 2 | 4.98 | 0.989 | **1s** |
| Greedy ensemble selection | 1161 | 4 | 3.84 | 0.975 | 2s |
| Linear model | 1168 | 3 | 3.81 | 0.975 | 5s |
| Nonlinear model | 1004 | 5 | 5.48 | 1.033 | 6s |
| Stacker model selection | 1158 | 3 | 3.87 | 0.967 | 295s |
| Multi-layer stacking | **1256** | **17** | **2.90** | **0.954** | 443s |

**Metrics**. For each dataset, we evaluate two tasks: point and probabilistic forecasting. Point accuracy is measured using *mean absolute scaled error* (MASE) on the median forecast, while probabilistic accuracy is assessed via *scaled quantile loss* (SQL) at quantile levels $\mathcal{Q} = \{0.1, 0.2, ..., 0.9\}$. To aggregate results across datasets, we report the *Elo rating*, number of wins (*champion*), *average rank*, and *average relative error* (geometric mean of errors normalized against the baseline). We use "Simple average (median)" as the baseline for Elo and relative error. Further details are provided in App. C.

**Setup**. We reserve the last $H$ observations of each time series as the test set. L1 models are trained using $K$=5-fold cross-validation, with the model from the last fold used for test prediction. Individual L2 models are trained on all 5 validation windows. For multi-layer stacking, the L3 model is fitted on the final window. After training, all models generate forecasts on the test set, and accuracy is evaluated using MASE or SQL, depending on the task.

**Results**. Complete results for 2 task types × 50 datasets × 44 models are reported in Tables 8–9 (appendix). To summarize these, we group forecast combination methods into 8 categories: simple average, model selection, performance-based average, greedy ensemble selection, linear model, nonlinear model, stacker model selection, and multi-layer stacking. We select the best-performing method (by average rank) from each category as its *representative method* and report aggregate metrics for these in Tables 1–2. Note that such aggregation does not give an unfair advantage to the multi-layer stacking approaches since there is only one method in each category. Moreover, multi-layer stacking has the best scores both in the full results (Tables 8–9) and in the aggregated results (Tables 1–2). Based on these results, we draw the following conclusions:

1. **Combination methods outperform individual forecasting models**. Combining multiple forecasts typically improves the accuracy compared to a single model, even if model selection is

performed. We see major accuracy gains, with up to 200 Elo points difference ($\approx 75\%$ win rate) and up to 5% error reduction. This result reaffirms the previous findings on the importance of ensembling in forecasting (X. Wang et al., 2023).

2. **Stacking outperforms simple aggregation techniques**. Contrary to the "forecast combination puzzle," which suggests that complex methods rarely outperform simple ones, our results clearly demonstrate that learned aggregation methods—like ensemble selection and linear models—yield substantially better accuracy than simple or performance-based averages.

3. **Multi-layer stacking outperforms individual combination methods**. Across both point and probabilistic tasks, multi-layer stacking achieves the highest accuracy. This result is consistent with our earlier observations: since no single L2 model performs best across all datasets, combining them in an L3 ensemble leads to improved overall accuracy. Notably, multi-layer stacking also outperforms model selection over L2 models, highlighting that combining multiple stackers is more effective than selecting the best one in isolation.

## 6.2 When and why does multi-layer stacking outperform other approaches?

**L3 model weights**. First, we investigate the weights assigned by the L3 ensemble selection algorithm to the underlying L2 models. We show the average weights in Fig. 3 and provide the per-dataset breakdown in Fig. 9–10 in the appendix. All constituent L2 stacker models receive non-zero weight, underscoring the value of maintaining a diverse portfolio. Notably, the nonlinear stacker LightGBM performs poorly on average when evaluated in isolation (Tab. 1), yet it is frequently selected by the L3 ensemble. This indicates that while LightGBM may underperform overall, it excels on specific datasets—which can be capitalized on by the adaptive multi-layer stacking framework.

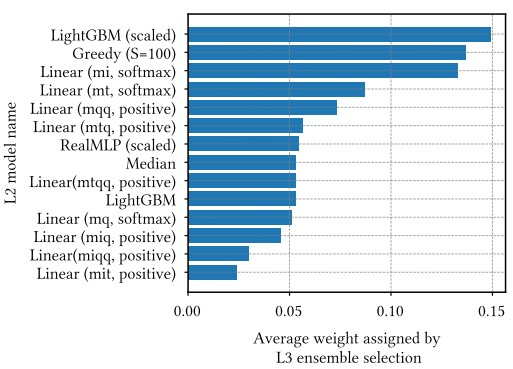

Figure 3: Weights assigned by the L3 ensemble selection algorithm to the L2 models (average over 50 tasks).

**Normalized performance**. To understand where multi-layer stacking succeeds or fails, we compare each representative model's score to the dataset-specific "champion" (Tab. 10). In 39 out of 50 datasets, most L2 models perform within 20% of the champion, and in these cases, multi-layer stacking typically ranks first or second in accuracy. In the remaining 11 datasets, where L2 model performance varies widely, multi-layer stacking tends to underperform. We hypothesize that allocating more validation data to train the L3 aggregator could mitigate this issue.

## 6.3 How much validation data do different combination methods require?

Previous experiments used 5 validation windows, requiring each base model to be trained 5 times. To reduce computational cost, we evaluated ensemble performance with fewer validation folds ($K = 1$ to $5$), using Median aggregation as a baseline since its performance is fold-independent.

Results show that even with just $K = 2$, multi-layer stacking outperforms all other methods, followed by the linear model. The nonlinear model benefits most from additional folds, while model selection performs best at $K = 1$, suggesting that older data is less indicative of the test set performance.

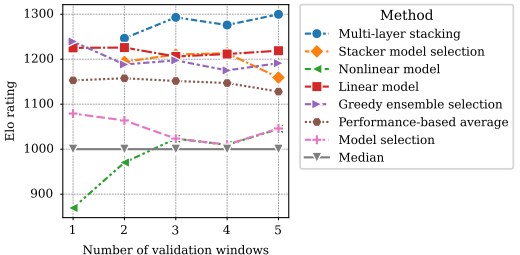

Figure 4: Influence of the number of validation windows on the model performance.

Table 3: Ablation: Aggregated probabilistic forecasting performance of representative combination methods when using only 6 base models. Best result in **bold**, second best underlined. Multi-layer stacking remains the top performer even with a reduced set of base models.

| Method | (↑) Elo | (↑) Champion | (↓) Average rank | (↓) Average relative error | (↓) Median marginal training time |
|---|---|---|---|---|---|
| Median | 1000 | 1 | 5.38 | 1.000 | **1s** |
| Model selection | 899 | 2 | 6.39 | 1.058 | **1s** |
| Performance-based average | 1095 | 7 | 4.46 | 0.994 | 2s |
| Greedy ensemble selection | 1110 | 6 | 4.23 | 0.990 | 7s |
| Linear model | 1145 | 2 | 3.93 | 0.978 | 8s |
| Nonlinear model | 1047 | 8 | 4.93 | 1.002 | 21s |
| Stacker model selection | 1128 | 2 | 4.07 | 0.982 | 418s |
| Multi-layer stacking | **1280** | **22** | **2.61** | **0.947** | 578s |

Table 4: Ablation: Aggregated probabilistic forecasting performance of multi-layer stacking with and without L2 model retraining. Scores are computed with respect to the methods in Tab. 1. Skipping retraining reduces fit time, but decreases forecast accuracy.

| Method | (↑) Elo | (↑) Champion | (↓) Average rank | (↓) Average relative error | (↓) Median marginal training time |
|---|---|---|---|---|---|
| Multi-layer stacking (L2 retraining) | **1306** | **20** | **2.81** | **0.945** | 720s |
| Multi-layer stacking (no L2 retraining) | 1253 | 13 | 3.24 | 0.950 | **483s** |

### 6.4 What is the effect of the L1 model choice?

Our experiments use a fixed set of 11 L1 forecasting models, but a robust ensembling method should perform well regardless of the base models used. To test this, we removed the 5 best-performing L1 models (*Chronos*, *PatchTST*, *TFT*, *DirectTabular*, *RecursiveTabular*) and retained only 6: *SeasonalNaive*, *AutoETS*, *Theta*, *DLinear*, *DeepAR*, and *TiDE*. The rest of the setup follows Sec. 6.1.

Aggregate results for 8 representative ensemble methods are shown in Tab. 3. The ranking remains unchanged: multi-layer stacking still performs best, showing robustness to L1 model choice. Model selection drops significantly due to the absence of Chronos (the strongest individual model), highlighting the importance of forecast combination when strong base models are unavailable.

### 6.5 Is it necessary to retrain the L2 models?

Our training procedure for the multi-layer stack ensemble (Sec. 4.2) includes a final step where L2 models are retrained on all validation windows. To assess whether this step is necessary, we perform an ablation comparing performance with and without this retraining. As shown in Tab. 4, skipping retraining leads to 1.5x faster training at the cost of lower forecast accuracy. This highlights the importance of retraining L2 models on the full validation data to ensure optimal performance.

## 7 Discussion

**Limitations & future work**. While our study demonstrates the effectiveness of learned forecast combination methods, several limitations remain. Most stacker models require predictions from all base models, which results in slower inference compared to sparse methods like ensemble selection. Future work could explore pruning strategy to mitigate this issue (Tsoumakas et al., 2009). Similarly, multi-layer stacking incurs substantial training costs, as we currently train 14 L2 models. A more principled approach—such as offline portfolio optimization, akin to Salinas and Erickson (2024)—could reduce computational overhead while maintaining or even improving accuracy. In addition, this framework can be extended with meta-learning approaches such as FFORMA (Montero-Manso et al., 2020), which aim to select or weight models based on dataset-level

features. Finally, the search for better aggregation models to use at the L2 and L3 levels remains an open and important challenge.

**Summary**. Our large-scale empirical study shows that learned ensembling methods—particularly stacking—consistently outperform both individual forecasting models and simple aggregation techniques. These findings challenge the "forecast combination puzzle" and demonstrate that flexible, data-driven ensemble strategies can significantly improve predictive accuracy. Multi-layer stacking extends this idea by combining multiple stackers, yielding robust performance across a wide range of datasets. While this approach provides consistent improvements overall, our ablations show it is especially valuable when strong individual forecasting models are unavailable. As forecasting tools and AutoML systems continue to evolve, we see strong potential in integrating advanced ensemble methods that adapt to data characteristics, scale efficiently, and generalize well across diverse forecasting scenarios.

## 8 Broader Impact Statement

After careful reflection, the authors have determined that this work presents no notable negative impacts to society or the environment.

**Acknowledgements**. We would like to thank David Holzmüller for his help with adding multi-quantile support to the RealMLP codebase, which enabled us to use this model in our evaluation.

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

# A Datasets

Table 5: Dataset statistics. Note that all datasets were filtered to only contain time series with at least $8 \times H$ observations to ensure that enough training & validation data is available.

| Dataset | Freq. | Seasonality | Horizon ($H$) | Num. Series | Num. Obs. | Min. Length | Max. Length |
|---|---|---|---|---|---|---|---|
| BDG-2 Bear | h | 24 | 48 | 91 | 1,482,312 | 8,760 | 17,544 |
| BDG-2 Bull | h | 24 | 48 | 41 | 719,304 | 17,544 | 17,544 |
| BDG-2 Fox | h | 24 | 48 | 135 | 2,324,568 | 8,760 | 17,544 |
| BDG-2 Hog | h | 24 | 48 | 24 | 421,056 | 17,544 | 17,544 |
| BDG-2 Panther | h | 24 | 48 | 105 | 919,800 | 8,760 | 8,760 |
| BDG-2 Rat | h | 24 | 48 | 280 | 4,728,288 | 8,760 | 17,544 |
| Beijing Air Quality | h | 24 | 48 | 132 | 4,628,448 | 35,064 | 35,064 |
| Beijing Subway | 30min | 48 | 96 | 552 | 867,744 | 1,572 | 1,572 |
| Borealis | h | 24 | 48 | 15 | 83,269 | 3,528 | 7,447 |
| CDC Fluview ILINet | W | 1 | 8 | 375 | 319,515 | 211 | 1,359 |
| Favorita Store Sales | D | 7 | 16 | 1,782 | 3,008,016 | 1,688 | 1,688 |
| Favorita Transactions | D | 7 | 16 | 54 | 91,152 | 1,688 | 1,688 |
| GEF12 | h | 24 | 48 | 11 | 433,554 | 39,414 | 39,414 |
| GEF17 | h | 24 | 48 | 8 | 140,352 | 17,544 | 17,544 |
| HZMetro | 15min | 96 | 96 | 160 | 380,320 | 2,377 | 2,377 |
| Hierarchical Sales | D | 7 | 14 | 118 | 212,164 | 1,798 | 1,798 |
| IDEAL | h | 24 | 48 | 217 | 1,255,253 | 393 | 16,167 |
| KDD Cup 2022 | 10min | 144 | 96 | 134 | 4,727,519 | 35,279 | 35,280 |
| Los-Loop | 5min | 288 | 96 | 207 | 7,094,304 | 34,272 | 34,272 |
| M-Dense | h | 24 | 48 | 30 | 525,600 | 17,520 | 17,520 |
| PEMS03 | 5min | 288 | 96 | 358 | 9,382,464 | 26,208 | 26,208 |
| PEMS08 | 5min | 288 | 96 | 510 | 9,106,560 | 17,856 | 17,856 |
| Project Tycho | W | 1 | 8 | 1,258 | 1,377,707 | 102 | 3,854 |
| SHMetro | 15min | 96 | 96 | 576 | 5,073,984 | 8,809 | 8,809 |
| SMART | h | 24 | 48 | 5 | 95,709 | 8,398 | 26,304 |
| SZ-Taxi | 15min | 96 | 96 | 156 | 464,256 | 2,976 | 2,976 |
| Subseasonal Precipitation | D | 7 | 14 | 862 | 9,760,426 | 11,323 | 11,323 |
| Australian Electricity | 30min | 48 | 96 | 5 | 1,155,264 | 230,736 | 232,272 |
| ERCOT | h | 24 | 24 | 8 | 1,238,976 | 154,872 | 154,872 |
| ETT (15 Min.) | 15min | 96 | 96 | 14 | 975,520 | 69,680 | 69,680 |
| ETT (Hourly) | h | 24 | 24 | 14 | 243,880 | 17,420 | 17,420 |
| Electricity (Hourly) | h | 24 | 48 | 321 | 8,443,584 | 26,304 | 26,304 |
| Electricity (Weekly) | W | 1 | 8 | 321 | 50,076 | 156 | 156 |
| FRED-MD | M | 12 | 12 | 107 | 77,896 | 728 | 728 |
| KDD Cup 2018 | h | 24 | 48 | 270 | 2,942,364 | 9,504 | 10,920 |
| M4 (Daily) | D | 1 | 14 | 4,218 | 10,022,845 | 112 | 9,933 |
| M4 (Hourly) | h | 24 | 48 | 414 | 373,372 | 748 | 1,008 |
| M4 (Monthly) | M | 12 | 18 | 32,436 | 9,773,903 | 144 | 2,812 |
| M4 (Quarterly) | Q | 4 | 8 | 19,049 | 2,167,663 | 64 | 874 |
| M4 (Weekly) | W | 1 | 13 | 294 | 365,534 | 260 | 2,610 |
| NN5 (Daily) | D | 7 | 56 | 111 | 87,801 | 791 | 791 |
| NN5 (Weekly) | W | 1 | 8 | 111 | 12,543 | 113 | 113 |
| Pedestrian Counts | h | 24 | 168 | 64 | 3,130,594 | 1,777 | 96,424 |
| Solar (10 Min.) | 10min | 144 | 96 | 137 | 7,200,720 | 52,560 | 52,560 |
| Taxi (30 Min.) | 30min | 48 | 96 | 2,428 | 3,589,798 | 1,469 | 1,488 |
| Taxi (Hourly) | h | 24 | 48 | 2,428 | 1,794,292 | 734 | 744 |
| Traffic (Weekly) | W | 1 | 8 | 862 | 89,648 | 104 | 104 |
| Uber TLC (Daily) | D | 7 | 14 | 262 | 47,422 | 181 | 181 |
| Uber TLC (Hourly) | h | 24 | 48 | 262 | 1,138,128 | 4,344 | 4,344 |
| Wind Farms (Daily) | D | 7 | 14 | 335 | 119,407 | 149 | 366 |

# B Models

## B.1 Base models

We considered the following base models, using their implementations provided by Auto-Gluon–TimeSeries (Shchur et al., 2023):

- `SeasonalNaive`: Simple model that sets the forecast equal to the last observed value from the same season (R. Hyndman and Athanasopoulos, 2021).

- `AutoETS`: Automatically tuned exponential smoothing with trend and seasonality (Garza et al., 2022; R. Hyndman, Koehler, et al., 2008; R. J. Hyndman and Khandakar, 2008)

- `DynamicOptimizedTheta`: A generalization of the Theta method by Assimakopoulos and Nikolopoulos (2000), that automatically selects and revises some of the model hyperparameters (Fiorucci et al., 2016).

- `DeepAR`: Autoregressive forecasting model based on a recurrent neural network (Salinas, Flunkert, et al., 2020).

- `PatchTST`: Transformer-based forecaster that segments each time series into patches (Nie et al., 2022).

- `TemporalFusionTransformer`: Combines LSTM with a transformer layer to predict the quantiles of all future target values (Lim et al., 2021).

- `DirectTabular`: Predict the future time series values by transforming the task into a tabular prediction task and then applying the LightGBM quantile regressor (Ke et al., 2017).

- `RecursiveTabular`: Predict the future time series values one-by-one by transforming the task into a tabular prediction task and then applying the LightGBM regressor (Ke et al., 2017). In contrast to `DirectTabular`, the forecast is computed step-by-step by repeatedly applying the tabular method.

- `TiDEModel`: Time series dense encoder model (Das et al., 2023).

- `Chronos`: Pretrained time series forecasting model (Ansari et al., 2024). We use the `bolt_base` configuration.

For simplicity, we do not investigate the effect of hyperparameter tuning on the models and keep all the hyperparameters to their default values. We do not expect this to affect our main conclusions, given the stability of our results with respect to the L1 model choice (Sec. 6.4).

For each dataset, we generate the base model predictions by fitting all models with 5 validation windows, refitting each model from scratch for each window. We set the maximum time limit of 30 minutes per window per model to avoid extremely long runtimes, though the vast majority of models complete training within 5 minutes per window. We save the base model predictions for all validation windows and the test window to disk. This enables us to train the stacker models without needing to re-fit the base models.

## B.2 Individual forecast combination methods

- **Simple averages**: We consider two simple averages, *mean* and *median*, which compute the mean or median of all base model predictions, respectively.

- **Model selection**: Model selection computes the index of the best-performing model according to the validation data during training, and then uses this individual model for forecasting.

- **Performance-weighted averages**: These models are weighted averages of the form $g(\hat{y}_1, \ldots, \hat{y}_M) = \sum_{m=1}^{M} \omega_m \hat{y}_m$, where the weights $\omega_m$ are computed directly from the (normalized) validation scores of the individual models, as $\omega_m \propto h(L_m)$, with $L_m \propto \mathcal{L}(\hat{y}_m, y_{\text{out}})$ such that $\sum_{m=1}^{M} L_m = 1$, and $\sum_{m=1}^{M} \omega_m = 1$. We consider three options for the function $h$:
  - Inv.: $h_{\text{inv}}(L) = 1/L$,
  - Sqr.: $h_{\text{sqr}}(L) = 1/L^2$,

– Exp.: $h_{\exp}(L) = \exp(1/L)$

Refer to Pawlikowski and Chorowska (2020) for more details.

- **Greedy ensembles**: This is the *ensemble selection* method by (Caruana et al., 2004), which optimizes the validation loss by greedily adding models to an equally-weighted ensemble *with replacement*. As an equally-weighted ensemble with replacement is equivalent to a weighted average with fractional weights, the greedy ensemble can also be interpreted as a weighted average $g(\hat{y}_1, \ldots, \hat{y}_M) = \sum_{m=1}^{M} \omega_m \hat{y}_m$ which is optimized via coordinate-wise ascent: Starting with zero weights $\omega^{(0)} = 0 \in \mathbb{R}^M$, the algorithm iterates for $j = 1, \ldots, S$, and selects at each step the model that would minimize the resulting loss, i.e.

$$m^{(j)} = \underset{m=1,\ldots M}{\arg\min} \, \mathcal{L}\left(g_{\hat{\omega}_m^{(j)}}(\hat{y}_1, \ldots, \hat{y}_M), y_{\text{ensemble}}\right), \tag{3}$$

where $\hat{\omega}_m^{(j)} := ((j-1)\omega^{(j-1)} + e_m)/j$ are the weights that the greedy ensemble would have if it were to select model $m$, with $e_m \in \mathbb{R}^M$ being a canonical basis vector, i.e. $[e_m]_k = \mathbb{1}_{k=m}$. It then adds the model to the ensemble and sets $\omega^{(j)} = \hat{\omega}_{m^{(j)}}^{(j)}$. This approach is currently also used as the default in multiple AutoML forecasting frameworks (Deng et al., 2022; Shchur et al., 2023; Zöller et al., 2024). In our evaluation, we consider three versions of this model, with the number of iterations $S$ set to 10, 100, and 1000, respectively.

- **Linear models**: For linear models, we consider a class of models $g_i$ of the form

$$[g_i(\hat{y}_{i,1}, \ldots, \hat{y}_{i,M})]_{h,q} = \sum_{m=1}^{M} \omega_{i,h,q,m} \cdot [\hat{y}_{i,m}]_{h,q} \tag{4}$$

with weights $\omega \in \mathbb{R}^{N \times H \times Q \times M}$. More concretely, we consider a number of variations of this model which satisfy different constraints or have different degrees of weight-tying:

– *Positivity and simplex constraints:* It is often desirable to enforce positivity and/or simplex constraints, that is, $\omega_{i,h,q,m} > 0$ and $\sum_m \omega_{i,h,q,m} = 1$, respectively (X. Wang et al., 2023). In our model, we enforce constraints by parameterizing the weights through an unconstrained weight $\tilde{\omega}_{i,h,q,m}$ and then passing it through an activation function. We consider two versions:
  * `softmax`: This version enforces both positivity and simplex constraints by passing the unconstrained weights through a softmax activation function:

$$\omega_{i,h,q,m} = \frac{\exp(\tilde{\omega}_{i,h,q,m})}{\sum_{m'} \exp(\tilde{\omega}_{i,h,q,m'})}. \tag{5}$$

  * `positive`: This version enforces positivity, but not the simplex constraint, by passing the unconstrained weights through a quadratic function:

$$\omega_{i,h,q,m} = \left(\tilde{\omega}_{i,h,q,m}\right)^2. \tag{6}$$

– *Weight-tying:* Instead of considering different weights for each model, item, time step, and quantile, we can lower the complexity by tying weights across some of these dimensions (Hasson et al., 2023). For example, tying weights across items would imply $\omega_{i,h,q,m} = \omega_{i',h,q,m}$ for all items $i, i'$. This yields a number of different linear models, denoted by the dimension along which the weights are *not* tied:
  * `m`: One weight per model; weights are tied across items, prediction times, and quantiles. This case is equivalent to weighted average models.
  * `mi`: One weight per model and per item; weights are tied across prediction times and quantiles.

* `mt`: One weight per model and per prediction time; weights are tied across items and quantiles.
* `mq`: One weight per model and per quantile; weights are tied across items and prediction times.
* `mit`: One weight per model, item, and prediction time; weights are tied across quantiles.
* `miq`: One weight per model, item, and quantile; weights are tied across prediction times.
* `mtq`: One weight per model, prediction time, and quantile; weights are tied across items.
* `mitq`: One weight per model, item, prediction time, and quantile; no weight-tying.

- *Across-quantile weights:* Instead of treating all quantiles independently, we can also compute quantile predictions *across quantiles* and rely on the base model estimates of all quantiles, as proposed by Fakoor et al. (2023); In its most flexible form, this corresponds to a model of the form

$$[g_i(\hat{y}_{i,1}, \ldots, \hat{y}_{i,M})]_{h,q} = \sum_{m=1}^{M} \sum_{q' \in \mathcal{Q}} \omega_{i,h,q,q',m} \cdot [\hat{y}_{i,m}]_{h,q'}, \tag{7}$$

with weights $\omega \in \mathbb{R}^{N \times H \times Q \times Q \times M}$. As before, we consider `softmax` and `positive` versions as well as different weight-tying options:

* `mqq`: One weight per model and across quantiles; weights are tied across items and prediction times.
* `miqq`: One weight per model, items, and across quantiles; weights are tied across prediction times.
* `mtqq`: One weight per model, prediction time, and across quantiles; weights are tied across items.

*Training.* We train linear stacker models by optimizing the resulting loss:

$$\omega = \underset{\omega' \in \mathbb{R}^{N \times H \times Q \times M}}{\arg\min} \sum_i \mathcal{L}\left(g_i(\hat{y}_{i,1}, \ldots, \hat{y}_{i,M}), y_{i,T+1:T+H}\right), \tag{8}$$

where $g_i$ is a linear model that uses the to-be-optimized weights $\omega'$. This can be approached with any numerical optimizer. In our implementation, we rely on the Adam optimizer (Kingma, 2014) and we use a custom learning rate schedule to reduce the learning rate whenever a plateau is hit and the loss starts oscillating. We also limit the training time to 10 minutes.

- **Nonlinear models**: To apply nonlinear models, we re-formulate the stacking problem as a tabular regression problem. Each item × timestep combination corresponds to a row in the training data. The quantile forecasts of the L1 models are used as features, and the ground truth time series value is the training target. We can then train any tabular regression model on this data. In our experiments we consider two methods:

- `RealMLP` is a deep-learning approach for tabular problems which improves on standard MLPs through a number of tricks and better, meta-learned default parameters (Holzmüller et al., 2024). We use the multi-quantile loss evaluated at levels $\mathcal{Q}$ as the training objective.
- `LightGBM` is a highly efficient implementation of gradient-boosted decision trees (Ke et al., 2017). We train a separate LightGBM regressor to predict each of the quantile levels $\mathcal{Q}$.

Additionally, we consider *scaled* versions of both methods in which we normalize the model predictions before feeding them into the tabular stacker model and then un-normalize the outputs again. This corresponds to a modified model $g'$ of the form

$$g'_i(\hat{y}_{i,1}, \ldots, \hat{y}_{i,M}) = \frac{g_i(\alpha \hat{y}_{i,1} + \beta, \ldots, \alpha \hat{y}_{i,M} + \beta) - \beta}{\alpha}, \tag{9}$$

where $\alpha, \beta$ are computed from the empirical mean and standard deviation of the predictions in order to standardize the inputs, and $g$ is a tabular stacker model as introduced above.

From each category we select one "representative model" for our condensed experiment summary in the paper, as described in Tab. 6 and Tab. 7.

Table 6: Representative models per category for the quantile forecast experiment.

| Model category | Model |
|---|---|
| Median | Median |
| Model Selection | Model selection |
| Performance-based average | Performance-based average (exp) |
| Greedy ensemble selection | Greedy (S=100) |
| Linear model | Linear(mq, softmax) |
| Nonlinear model | LightGBM (scaled) |

Table 7: Representative models per category for the point forecast experiment.

| Model category | Model |
|---|---|
| Median | Median |
| Model Selection | Model selection |
| Performance-based average | Performance-based average (exp) |
| Greedy ensemble selection | Greedy (S=100) |
| Linear model | Linear(m, softmax) |
| Nonlinear model | LightGBM (scaled) |

### B.3 Multi-layer stacker models

For multilayer stacking, we use the following 14 stacker models as L2 models:

- Median
- Greedy (S=100)
- Linear (mi, softmax)
- Linear (mt, softmax)
- Linear (mq, softmax)
- Linear (mit, positive)
- Linear (mtq, positive)
- Linear (miq, positive)
- Linear (mqq, positive)
- Linear (miqq, positive)
- Linear (mtqq, positive)
- LightGBM
- LightGBM (scaled)
- RealMLP (scaled)

We have arbitrarily chosen these 14 L2 models since they provide a good coverage of different model families. To ensure a fair comparison, we have fixed this selection before running any experiments, and did not adjust the selection to optimize the benchmark performance.

We then consider multi-layer stackers resulting from two different L3 models:

- **Stacker model selection** uses *model selection* as its L3 model and thus relies on a single L2 model during test time.

- **Multi-layer stacking** uses *Greedy (S=100)* as its L3 model to compute a weighted average of the L2 model predictions.

## C Evaluation metrics

### C.1 Loss functions

Loss functions are used both to train and evaluate time series models. In Sec. 2.1, we introduced them as functions that take in the (quantile) prediction of a particular item as well as the ground-truth values and return a positive scalar value, that is

$$\mathcal{L} : \mathbb{R}^{H \times Q} \times \mathbb{R}^H \to \mathbb{R}_{\geq 0}, \tag{10}$$

with lower values indicating a more accurate forecast. We consider the following two losses to evaluate quantile forecasts and point forecasts, respectively:

- **Scaled quantile loss (SQL):**

$$\text{SQL}(\hat{y}_i, y_i) := \frac{1}{H} \frac{1}{Q} \frac{1}{a_i} \sum_{h=1}^{H} \sum_{q \in \mathcal{Q}} \rho_q \left( \hat{y}_{i,T+h}^q, y_{i,T+h} \right), \tag{11}$$

where $\rho_q : \mathbb{R} \times \mathbb{R} \to \mathbb{R}$ is the *quantile loss at level q*, defined as

$$\rho_q \left( \hat{y}_{i,h}^q, y_{i,h} \right) := 2 \cdot \begin{cases} q \cdot \left( y_{i,h} - \hat{y}_{i,h}^q \right), & \text{if } y_{i,h} < \hat{y}_{i,h}^q, \\ (1 - q) \cdot \left( \hat{y}_{i,h}^q - y_{i,h} \right), & \text{if } y_{i,h} \geq \hat{y}_{i,h}^q, \end{cases} \tag{12}$$

and where $a_i$ is the *historic absolute seasonal error* of the time series, defined as

$$a_i = \frac{1}{T - m} \sum_{t=m+1}^{T} \left| y_{i,t} - y_{i,t-m} \right|, \tag{13}$$

with $m$ the seasonality of the dataset provided in Tab. 5.

- **Mean absolute scaled error (MASE):**

$$\text{MASE}(\hat{y}_i, y_i) := \frac{1}{H} \frac{1}{a_i} \sum_{h=1}^{H} \left| \hat{y}_{i,T+h} - y_{i,T+h} \right|, \tag{14}$$

where $a_i$ is the historic absolute seasonal error, as defined above. Note that MASE is equivalent to an SQL applied only to the 0.5 quantile.

To evaluate a particular time series model on a whole dataset, we average the loss across all individual items.

When evaluating point and probabilistic forecast performance, we used MASE or SQL as the training objective, respectively. This means, that all L2 and L3 models were trained once for point forecasting tasks using MASE as the training loss, and one more time for probabilistic forecasting tasks using the SQL loss.

### C.2 Result aggregation

To summarize the empirical evaluation, we aggregate the individual losses $L_{m,d}$ of each model $m$ and dataset $d$ in a number of ways:

- **Elo**: The Elo rating system was originally introduced to calculate the relative skill levels of players in zero-sum games such as chess (Elo, 1967), but it has recently also been used to evaluate large language models (Bai et al., 2022; Boubdir et al., 2024). We compute our Elo scores as done in the recently proposed "Chatbot Arena" (Chiang et al., 2024), and calibrate the computation such that our chosen baseline has an Elo of 1000.

- **Average rank**: First, for each dataset in our evaluation, we rank the methods by their achieved error (resolving ties such that models receive the average rank of the tied models). Then, we compute the average rank of each model over all datasets with an arithmetic mean.

- **Champion**: This metric counts for how many datasets the method has achieved the lowest error among all methods included in the comparison. As we already computed a ranking for each dataset, this can be done by simply counting how often a method achieves rank 1. Since sometimes multiple methods are tied for the first place, the sum of the values in the "Champion" column does not always add up to the number of the datasets.

- **Average relative error**: To make the errors on different datasets more comparable, we first compute *relative* errors with respect to a chosen baseline method $\overline{m}$ (namely the "simple average (median)" method), as

$$L_{m,d}^{\text{rel}} := \frac{L_{m,d}}{L_{\overline{m},d}}. \tag{15}$$

To limit the influence of outliers, we clip the individual relative errors to the range $[10^{-3}, 5]$. Then, we compute the *average relative error* of each model $m$ across all datasets by aggregating the respective relative errors with a geometric mean:

$$\text{AverageRelativeError}_m = \text{GeometricMean}\left(L_{m,1}^{\text{rel}}, \ldots, L_{m,D}^{\text{rel}}\right). \tag{16}$$

## D  Computational resources

All models were trained on cloud-hosted machines with 16 vCPUs and 64GB RAM. The base models were trained with a time limit of 30 minutes per window per model to avoid extremely long runtimes, though the vast majority of models complete training within 5 minutes per window. The inidividual L2 stacker models were trained as described in App. B.2, either for a fixed, pre-specified number of steps, or until convergence within a time limit of 10 minutes. See also Tab. 11 and Tab. 12 for the median training times of the base and stacker models.

## E  Additional figures and tables

- Tab. 8: Full probabilistic forecasting results. SQL error values for all datasets × all methods (11 base models and 33 combination methods).

- Tab. 9: Full point forecasting results. MASE error values for all datasets × all methods (11 base models and 33 combination methods).

- Fig. 5: Critical differences (CD) diagram for 8 representative combination methods (probabilistic forecasting / SQL).

- Fig. 6: Critical differences (CD) diagram for 8 representative combination methods (point forecasting / MASE).

- Fig. 7: Distribution of relative error scores for 8 representative combination methods (probabilistic forecasting / SQL).

- Fig. 8: Distribution of relative error scores for 8 representative combination methods (point forecasting / MASE).

- Tab. 11: Median training time per validation window for L1 models.

- Tab. 12: Median end-to-end training time for the ensemble models.

- Fig. 9: Weights assigned by the L3 ensemble selection algorithm to the constituent L2 stacker models (probabilistic forecasting / SQL).

- Fig. 10: Weights assigned by the L3 ensemble selection algorithm to the constituent L2 stacker models (point forecasting / MASE).

- Tab. 10: Normalized error for each representative combination method and dataset combination (probabilistic forecasting / SQL).

### E.1 Critical differences diagrams

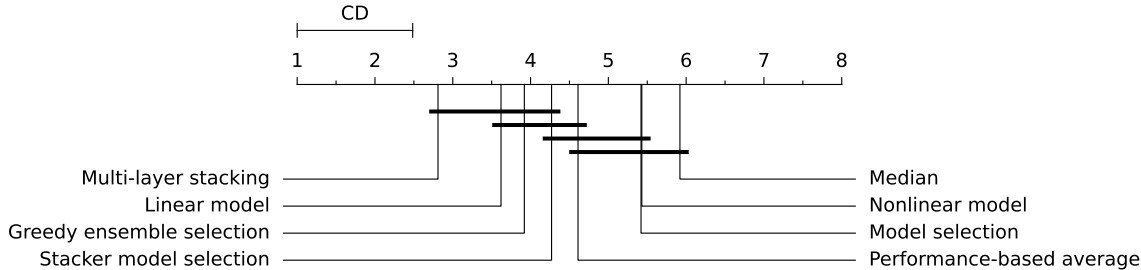

Figure 5: Critical differences (CD) diagram for 8 representative combination methods. Based on probabilistic forecasting tasks (SQL metric).

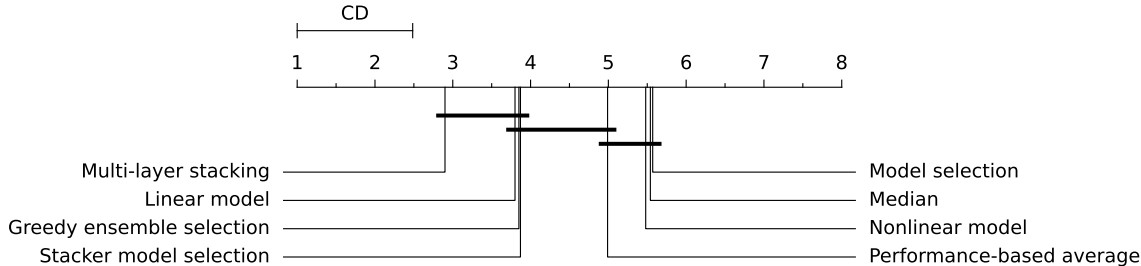

Figure 6: Critical differences (CD) diagram for 8 representative combination methods. Based on point forecasting tasks (MASE metric).

## E.2 Distribution of relative errors

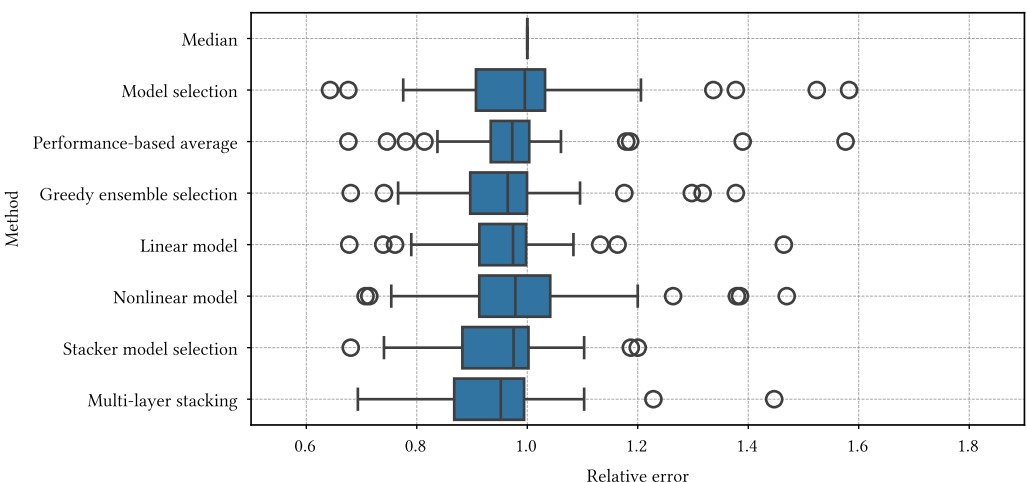

Figure 7: Distribution of the relative errors for each combination method (normalized by the performance of median aggregation). Based on probabilistic forecasting tasks (SQL metric).

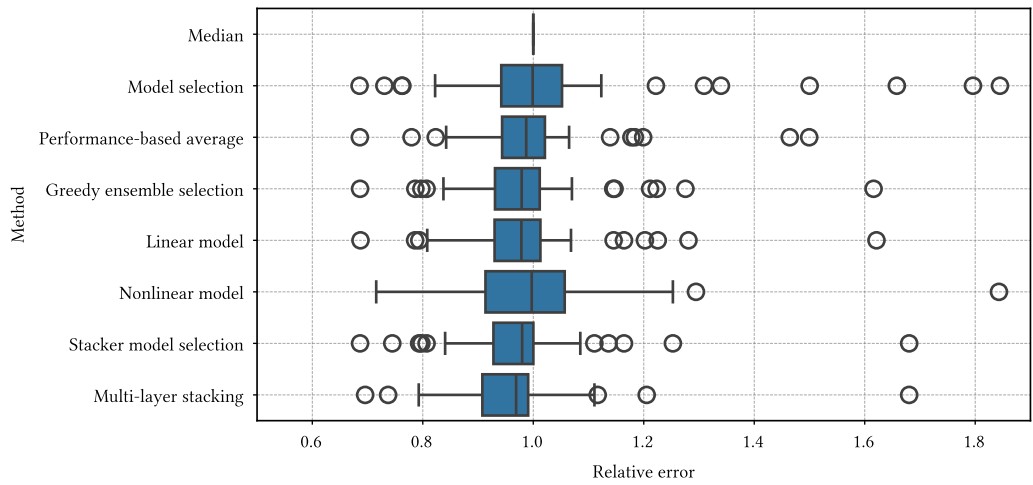

Figure 8: Distribution of the relative errors for each combination method (normalized by the performance of median aggregation). Based on point forecasting tasks (MASE metric).

Table 8: Quantile forecast accuracy for base and stacker models, measured with scaled quantile loss (SQL). Lower values are better.

Table 9: Point forecast accuracy for base and stacker models, measured with mean absolute scaled error (MASE). Lower values are better.

| Dataset | Individual models | | | | | | | | | | | Simple avg. | | Model sel. | Performance-weighted | | | Greedy ensemble | | | Linear-softmax | | | | | | | | | Linear-positive | | | | | | | | | Nonlinear models | | | | Multi-layer | |
|---|---|---|---|---|---|---|---|---|---|---|---|---|---|---|---|---|---|---|---|---|---|---|---|---|---|---|---|---|---|---|---|---|---|---|---|---|---|---|---|---|---|---|---|
|  | AutoETS | Theta | SeasonalNaive | PatchTST | DeepAR | DLinear | TiDE | DirectTabular | RecursiveTabular | TFT | Chronos-Bolt | Mean | Median | Model Selection | Exp. | Inv. | Sqr. | S=10 | S=100 | S=1000 | m | mi | miq | mit | mitq | mq | mqq | mt | mtq | m | mi | miq | mit | mitq | mq | mqq | mt | mtq | RealMLP | RealMLP (scaled) | LightGBM | LightGBM (scaled) | Stacker Model Selection | Multi-layer Stacking |
| Average rank | 19.030 | 19.080 | 20.970 | 33.200 | 22.410 | 32.180 | 31.220 | 32.230 | 33.560 | 32.620 | 21.000 | 30.180 | 19.670 | 21.000 | 17.050 | 24.130 | 21.410 | 13.030 | 12.590 | 12.830 | 12.760 | 17.330 | 17.330 | 29.580 | 29.580 | 12.760 | 13.300 | 16.870 | 16.870 | 12.760 | 17.330 | 17.330 | 29.580 | 29.580 | 12.760 | 13.300 | 16.870 | 16.870 | 10.380 | 21.290 | 23.930 | 19.900 | 12.870 | 9.620 |
| Gmean relative error | 1.737 | 1.533 | 1.349 | 1.192 | 1.177 | 1.213 | 1.110 | 1.275 | 1.154 | 1.173 | 1.011 | 1.105 | 1.000 | 1.026 | 0.989 | 1.027 | 1.007 | 0.975 | 0.975 | 0.975 | 0.975 | 0.997 | 0.997 | 1.064 | 1.064 | 0.975 | 0.973 | 0.992 | 0.992 | 0.975 | 0.997 | 0.997 | 1.064 | 1.064 | 0.975 | 0.973 | 0.992 | 0.992 | 1.315 | 1.033 | 1.082 | 1.033 | 0.967 | 0.954 |

Datasets (rows): BEIJING-SUBWAY-30MIN, ETTh, ETTm, HZMETRO, LOS-LOOP, M-DENSE, PEMS03, PEMS08, SHMETRO, SZ-TAXI, australian-electricity, bdg-2-bear, bdg-2-fox, bdg-2-panther, bdg-2-rat, beijing-air-quality, borealis, bull, cdc-fluview-ilinet, electricity-hourly, electricity-weekly, ercot, fred-md, gfc12-load, gfc17-load, hierarchical-sales, hog, ideal, kdd-cup-2018, kdd2022, m4-daily, m4-hourly, m4-monthly, m4-quarterly, m4-weekly, m5, m5-weekly, pedestrian-counts, project-tycho, smart, solar-10-minutes, store-sales, subseasonal-precip, taxi-1h, taxi-30min, traffic-weekly, transactions, uber-tlc-daily, uber-tlc-hourly, wind-farms-daily

Table 10: Normalized error of each representative combination method per dataset, relative to the best model for that dataset (probabilstic tasks / SQL metric). Two key observations: (1) When most stackers perform reasonably (normalized error < 1.2), multi-layer stacking typically ranks among the top performers. (2) It underperforms when most constituent stackers—especially nonlinear ones—produce poor forecasts.

| Dataset | Median | Model selection | Performance-based average | Greedy ensemble selection | Linear model | Nonlinear model | Stacker model selection | Multi-layer stacking |
|---|---|---|---|---|---|---|---|---|
| ETTh | 1.068 | 1.173 | 1.049 | 1.056 | 1.050 | 1.157 | 1.025 | 1.000 |
| HZMETRO | 1.096 | 1.139 | 1.115 | 1.095 | 1.086 | 1.029 | 1.032 | 1.000 |
| LOS-LOOP | 1.364 | 1.301 | 1.146 | 1.104 | 1.095 | 1.046 | 1.056 | 1.000 |
| PEMS08 | 1.378 | 1.124 | 1.028 | 1.056 | 1.048 | 1.039 | 1.039 | 1.000 |
| bdg-2-bear | 1.089 | 1.107 | 1.064 | 1.087 | 1.123 | 1.070 | 1.045 | 1.000 |
| bdg-2-rat | 1.240 | 1.051 | 1.067 | 1.051 | 1.038 | 1.055 | 1.016 | 1.000 |
| borealis | 1.090 | 1.068 | 1.051 | 1.047 | 1.055 | 1.014 | 1.017 | 1.000 |
| hog | 1.199 | 1.224 | 1.146 | 1.160 | 1.120 | 1.040 | 1.040 | 1.000 |
| ideal | 1.041 | 1.021 | 1.007 | 1.001 | 1.001 | 1.006 | 1.010 | 1.000 |
| m4-daily | 1.007 | 1.042 | 1.029 | 1.029 | 1.005 | 1.479 | 1.007 | 1.000 |
| m4-hourly | 1.350 | 1.149 | 1.149 | 1.101 | 1.089 | 1.029 | 1.027 | 1.000 |
| m4-monthly | 1.043 | 1.436 | 1.097 | 1.436 | 1.078 | 2.118 | 1.024 | 1.000 |
| m4-quarterly | 1.033 | 1.066 | 1.029 | 1.017 | 1.019 | 1.001 | 1.012 | 1.000 |
| m4-weekly | 1.173 | 1.250 | 1.098 | 1.084 | 1.073 | 1.109 | 1.042 | 1.000 |
| nn5 | 1.057 | 1.025 | 1.008 | 1.007 | 1.008 | 1.021 | 1.008 | 1.000 |
| nn5-weekly | 1.043 | 1.046 | 1.017 | 1.025 | 1.026 | 1.083 | 1.025 | 1.000 |
| project-tycho | 1.020 | 1.011 | 1.025 | 1.004 | 1.001 | 1.003 | 1.001 | 1.000 |
| store-sales | 1.022 | 1.000 | 1.013 | 1.001 | 1.007 | 1.012 | 1.007 | 1.000 |
| transactions | 1.124 | 1.103 | 1.063 | 1.049 | 1.048 | 1.047 | 1.015 | 1.000 |
| uber-tlc-daily | 1.027 | 1.024 | 1.002 | 1.001 | 1.009 | 1.026 | 1.027 | 1.000 |
| SZ-TAXI | 1.027 | 1.055 | 1.014 | 1.001 | 1.000 | 1.011 | 1.001 | 1.001 |
| PEMS03 | 1.076 | 1.000 | 1.067 | 1.064 | 1.049 | 1.108 | 1.076 | 1.005 |
| kdd-cup-2018 | 1.163 | 1.046 | 1.026 | 1.002 | 1.000 | 1.092 | 1.000 | 1.005 |
| hierarchical-sales | 1.011 | 1.009 | 1.013 | 1.003 | 1.000 | 1.037 | 1.000 | 1.005 |
| taxi-30min | 1.188 | 1.002 | 1.003 | 1.002 | 1.000 | 1.008 | 1.008 | 1.005 |
| bull | 1.008 | 1.031 | 1.000 | 1.017 | 1.005 | 1.048 | 1.017 | 1.008 |
| uber-tlc-hourly | 1.151 | 1.000 | 1.003 | 1.000 | 1.001 | 1.016 | 1.013 | 1.010 |
| pedestrian-counts | 1.026 | 1.060 | 1.060 | 1.052 | 1.039 | 1.000 | 1.028 | 1.011 |
| ercot | 1.121 | 1.124 | 1.052 | 1.000 | 1.069 | 1.030 | 1.000 | 1.016 |
| traffic-weekly | 1.043 | 1.084 | 1.020 | 1.000 | 1.004 | 1.064 | 1.045 | 1.025 |
| taxi-1h | 1.479 | 1.000 | 1.000 | 1.006 | 1.003 | 1.046 | 1.006 | 1.026 |
| smart | 1.030 | 1.054 | 1.000 | 1.032 | 1.032 | 1.121 | 1.032 | 1.027 |
| ETTm | 1.124 | 1.000 | 1.052 | 1.037 | 1.026 | 1.230 | 1.042 | 1.027 |
| electricity-weekly | 1.290 | 1.000 | 1.050 | 1.017 | 1.041 | 1.175 | 1.017 | 1.030 |
| beijing-air-quality | 1.343 | 1.042 | 1.048 | 1.089 | 1.061 | 1.093 | 1.000 | 1.030 |
| wind-farms-daily | 1.030 | 1.000 | 1.040 | 1.005 | 1.011 | 1.074 | 1.074 | 1.036 |
| subseasonal-precip | 1.151 | 1.127 | 1.126 | 1.078 | 1.058 | 1.000 | 1.053 | 1.037 |
| electricity-hourly | 1.245 | 1.000 | 1.043 | 1.040 | 1.046 | 1.090 | 1.040 | 1.037 |
| cdc-fluview-ilinet | 1.072 | 1.077 | 1.000 | 1.015 | 1.027 | 1.201 | 1.110 | 1.048 |
| BEIJING-SUBWAY-30MIN | 1.055 | 1.607 | 1.466 | 1.072 | 1.073 | 1.000 | 1.090 | 1.051 |
| bdg-2-panther | 1.173 | 1.000 | 1.129 | 1.069 | 1.082 | 1.157 | 1.144 | 1.064 |
| fred-md | 1.000 | 1.127 | 1.011 | 1.015 | 1.000 | 1.009 | 1.088 | 1.068 |
| SHMETRO | 1.000 | 1.018 | 1.017 | 1.022 | 1.015 | 1.094 | 1.094 | 1.075 |
| kdd2022 | 1.034 | 1.382 | 1.226 | 1.341 | 1.120 | 1.000 | 1.140 | 1.140 |
| gfc12-load | 1.077 | 1.299 | 1.023 | 1.000 | 1.060 | 1.361 | 1.133 | 1.140 |
| bdg-2-fox | 1.555 | 1.000 | 1.335 | 1.152 | 1.150 | 1.111 | 1.152 | 1.150 |
| M-DENSE | 1.058 | 1.052 | 1.009 | 1.000 | 1.071 | 1.270 | 1.270 | 1.159 |
| australian-electricity | 1.076 | 1.000 | 1.141 | 1.264 | 1.218 | 1.484 | 1.277 | 1.321 |
| gfc17-load | 1.000 | 2.568 | 1.179 | 1.317 | 1.464 | 1.385 | 1.991 | 1.447 |
| solar-10-minutes | 1.000 | 1.583 | 1.576 | 1.096 | 1.163 | 2.945 | 2.945 | 2.158 |

Table 11: Median training time for L1 models per one validation window (in seconds) for probabilistic tasks.

| Model | Duration (seconds) |
|---|---|
| AutoETS | 14 |
| Chronos | 8 |
| DLinear | 26 |
| DeepAR | 149 |
| DirectTabular | 233 |
| Theta | 8 |
| PatchTST | 63 |
| RecursiveTabular | 51 |
| SeasonalNaive | 1 |
| TFT | 204 |
| TiDE | 176 |

Table 12: Median end-to-end training time for ensemble models (in seconds) for probabilistic tasks.

| Model | Duration (seconds) |
|---|---|
| Simple Average (Mean) | 1 |
| Simple Average (Median) | 1 |
| Model Selection | 1 |
| Performance-based (Exp.) | 2 |
| Performance-based (Inv.) | 2 |
| Performance-based (Sqr.) | 2 |
| Greed (S=10) | 2 |
| Greedy (S=100) | 11 |
| Greed (S=1000) | 70 |
| Linear (m, softmax) | 15 |
| Linear (m, positive) | 13 |
| Linear (mi, softmax) | 7 |
| Linear (mi, positive) | 34 |
| Linear (miq, softmax) | 12 |
| Linear (miq, positive) | 44 |
| Linear (mit, softmax) | 12 |
| Linear (mit, positive) | 44 |
| Linear (mitq, softmax) | 30 |
| Linear (mitq, positive) | 77 |
| Linear (mq, softmax) | 11 |
| Linear (mq, positive) | 15 |
| Linear (mqq, softmax) | 10 |
| Linear (mqq, positive) | 33 |
| Linear (mt, softmax) | 7 |
| Linear (mt, positive) | 31 |
| Linear (mtq, softmax) | 13 |
| Linear (mtq, positive) | 52 |
| LightGBM | 25 |
| LightGBM (scaled) | 26 |
| RealMLP | 200 |
| RealMLP (scaled) | 207 |
| Stacker Model Selection | 484 |
| Multi-layer Stacking | 721 |

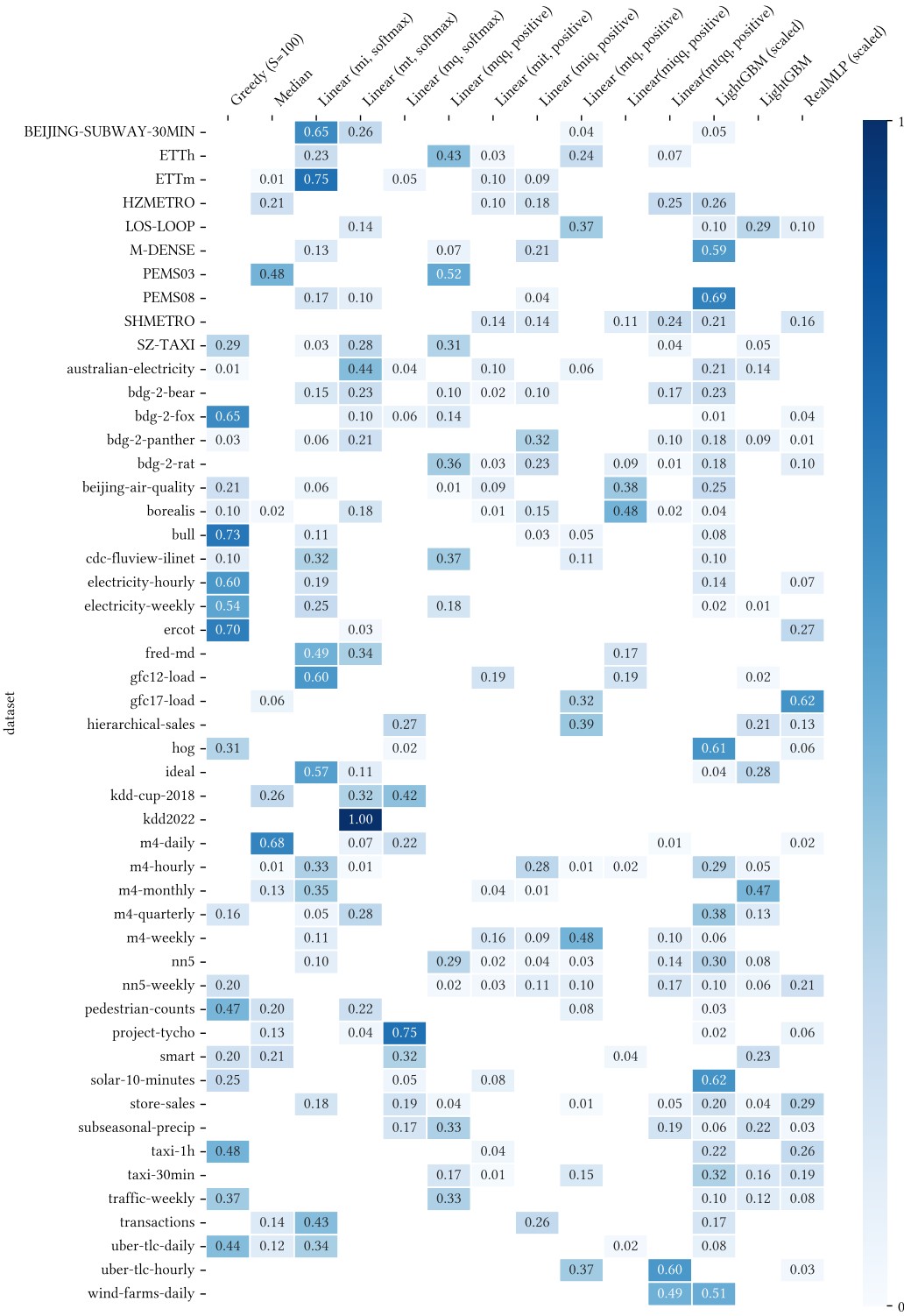

Figure 9: Weights assigned by the L3 ensemble selection algorithm to the L2 stacker models over 50 probabilistic forecasting tasks.

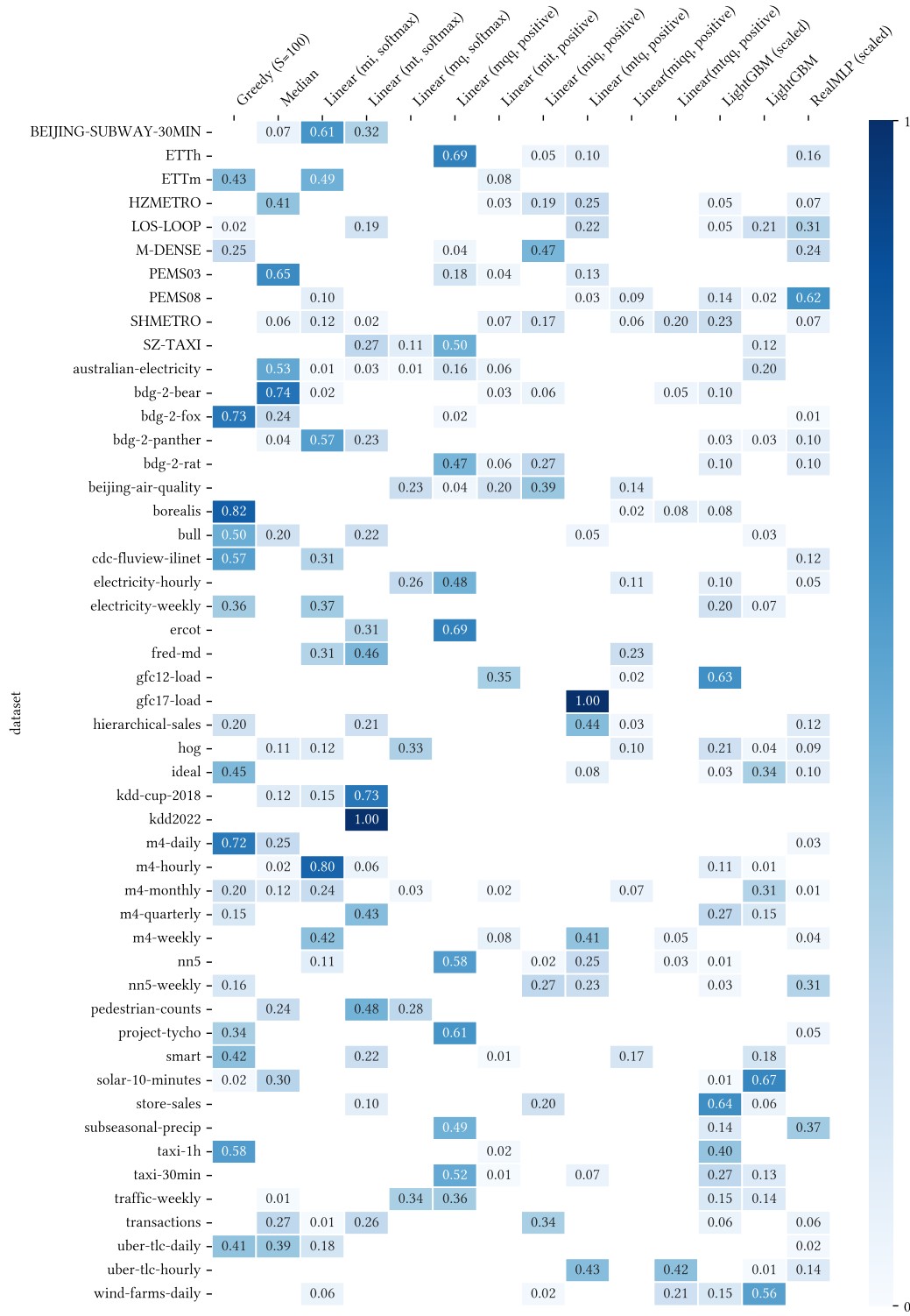

Figure 10: Weights assigned by the L3 ensemble selection algorithm to the L2 stacker models over 50 point forecasting tasks.

