# OpenReview forum: "Multi-layer Stack Ensembles for Time Series Forecasting"
_automl.cc/AutoML/2025/Methods_Track — AutoML 2025 Methods Track_

### Official Review · Reviewer_GgWT · 2025-04-05

**Comments To Authors:**

Strengths -
About 33 benchmarked forecast combination methods, been done by author across 50 datasets.
three-layer stacking architecture strengthens the manuscript

Weakness -
Inclusion of some statistical significance tests (e.g., Wilcoxon signed-rank) would help to strengthen robustness
It is observed that a fixed L2 Model Set in Multi-layer Stacking is used in the manuscript
Literature review and references used are shallow and can be further improved to make them in depth
Paper does not provide Evaluation on Multivariate Forecasting

**Review Confidence:**

4

**Review Rating:**

6

---

### Official Review · Reviewer_PFnc · 2025-04-28

**Comments To Authors:**

This paper presents a multi-layer stacking method for time series forecasting, porting existing advances done in tabular data to the area of univariate time series forecasting. The paper is well written, easy to read, understand and follow, and the contributions are solid. I like it and I think it's a valuable paper. I specially liked its state of the art and evaluation, which are very comprehensive, making it easy for a reader to understand the field and get a sense of the current best performing methods.

On the other hand, I was surprised by several decisions, which lowered my overall score of the paper and that I detail below:

1. The main one is that I have mixed feelings about how the evaluation is carried out and analyzed, as well as the advice given to use the method in practice.
	1. First of all, Table 1 shows that the linear model has almost equal average relative error as the multi-layer stacking. While the Elo rating show that the stacking method is, indeed, better than the rest, I think this fact shows the need for a deeper analysis, not just one made with victors and losers. Given that the linear method is 70x faster than the multi-layer stacking, it should be clear (and in the paper, not the Appendix) when one should decide to go for one or the other. This is, supposedly, answered in section 6.2, but in practice there's no real indication on when one should use a linear model or go for a more complex model.
	2. A contributing factor to the deeper insights to be extracted I mention above comes from the absence of uncertainty quantification in any of the results shown: no result shown displays confidence intervals of any matter, which makes understanding the results much harder. While they are displayed in the Appendices, that information shouldn't be left for it.
	3. Lastly, it feels strange to read in the Submission Checklist that:
		1. The authors didn't repeat the experiments with multiple seeds due to a large number of evaluated models and datasets. Based on the given numbers, a whole run over the 50 datasets of the multi-layer stacking would take around 10 hours. Assuming that all the others together take a bit more than that, we are at an estimation of ~1 day for a complete run. If this is the case, I find it strange that the authors couldn't run the experiments again with different seeds.
		2. The authors state that they have not run statistical tests... but then appendix E.1 shows two critical differences diagrams, based around statistical tests, in which the proposed method can't be assumed to be significantly better than a linear model. But this is probably due to the CD being too wide, as K = 8. While I don't think the Elo approach is a bad one, I'd have appreciated complementing it with one that also takes into account the actual difference between methods, specially considering how close the linear model seems to be to the multi-layer stacking.
2. In section 4.1, Fig. 2, why was it decided to train L3 with just one partition? I don't know why it couldn't be varied, training L2 with K blocks and L3 with T blocks, with K+T = |L1-pred|. This is specially important when the authors state, in section 6.2, that allocating more validation data to the L3 aggregator could mitigate an observed issue. Why wasn't this done already?
3. As a smaller comment, the number of datasets in Table 1 (summing the Champion column) sums to 47, while in Table 2 it sums to 49, whereas Table 5 has 50 datasets. Why this discrepancy?

Overall, I was slightly disappointed by the problems I outline, because I do like the paper and I think the contributions are valuable, but I felt that the evaluation and practical advice to use it could and should be improved.

**Review Confidence:**

4

**Review Rating:**

5

---

### Official Review · Reviewer_Hiuv · 2025-04-29

**Comments To Authors:**

This paper deals with the important problem of time series ensembling, which aims to combine estimates from different models to improve performance. Stacking is a machine learning concept that aims to combine base learners into a final learner by learning weights to maximize performance on held-out data. The authors note that while different stacking methods exist, there is no clear consensus on which method is most useful, because previous studies have shown that the optimal stacking technique depends strongly on the dataset.  The authors propose what they call “multi-layer stacking” in which a class of stacker models are each applied to the dataset. Then, an aggregator is learned which combines the outputs from each stacker model. In the “multi-layer stacking” approach, there are then two sets of parameters to learn (those that combine base models into stacker models, and those that combine stacker models into the aggregator), whereas in the usual stacking method, there is just one set of parameters (those that combine base models into the aggregator). The paper considers a wide range of datasets from different domains  and shows the performance of the “multi-layer stacking” method with various metrics. The proposed method tends to have the highest performance when compared to other standard methods (with some caveats that I discuss below). Overall, I think the paper's contributions are interesting and provide a solution to the issue of deciding which stacker model to use in time-series predictions.

I am not an expert in the literature in this area, so I leave this to other reviewers to comment on the thoroughness of the literature review.

Questions and concerns:
1. I think there should be more discussion on the differences between the proposed “multi-layer stacking” method and a neural network with 2 hidden layers, which seems very similar (at least for certain cases of stacking models and base learners).
2. I’m unclear on the average relative error column interpretation. Does an average error of 1 mean that the |prediction - truth|/|truth| = 1? In other words, that the prediction is one order of magnitude off from the target? Can you please clarify the interpretation of this column.
3.  Another concern is about training time needed to train the multi-layer stacking method. Table 1, for example, shows that the proposed method takes 721 seconds to train, whereas competing methods take several seconds. While these differences are quite larger, the training time for the proposed method still finishes in under 15 minutes, so it’s reasonably fast to train. But I think the paper could benefit from more emphasis being placed on the training time, as well as possible solutions to reduce training time. In Section 7 the authors do mention ways that training time could be reduced, but it could be beneficial to discuss this more, especially given the large differences in training time between stacking and multi-layer stacking.

**Review Confidence:**

2

**Review Rating:**

7

---

### Meta-Review · Area_Chair_WRCS · 2025-05-11

**Recommendation:** Accept
**Confidence:** 3

**Metareview:**

The paper is clear, addresses a relevant problem and proposes an interesting approach.

The main limitations are some questions about the evaluation and the lack of a very relevant survey in the citations: Wang, X., Hyndman, R. J., Li, F., & Kang, Y. (2023). Forecast combinations: An over 50-year review. International Journal of Forecasting, 39(4), 1518–1547. https://doi.org/10.1016/j.ijforecast.2022.11.005

Please note that one of the reviewers requests a large number of papers to be cited and it isn't clear to me the relevance of most of them to this work. Therefore, I assume this was a mistake from the reviewer and ask the authors to carefully check which (if any) of those papers are relevant and only cite the ones that you deem relevant to your work.